# Position: Good Embodied Reward Models Need Bad Behavior Data

Ran "Thomas" Tian [1 2]   Yilin Wu [3]   Andrea Bajcsy [3]

## Abstract

This position paper argues that to obtain reliable embodied reward models, the community must invest in "bad" robot data: failed, suboptimal, error-prone, and even hazardous behaviors. While reward models are central to any foundation model's lifecycle, today's embodied reward models are trained primarily on successful behaviors. We analyze three state-of-the-art embodied reward models and find that they systematically over-reward behaviors that real human evaluators would penalize, including unsafe interactions, poor execution, and shortcut strategies that only superficially satisfy tasks. We attribute these failures to a key data gap: the scarcity of negative embodied data which is costly to collect and often filtered out or withheld in existing robotics datasets. Furthermore, we show that even modest exposure to real bad behavior data can improve alignment with human preferences and reduce costly false positives. We therefore call on the embodied AI community to curate and release their bad robot data, build synthetic bad data generation engines, develop more decentralized physical evaluation systems, and design benchmarks for fine-grained embodied reward model evaluations.

## 1. Introduction

Reward models are central to the lifecycle of any foundation model, from reinforcement learning-based post-training (Ouyang et al., 2022), to test-time compute (Snell et al., 2024), to large-scale evaluation (Wang et al., 2023). For example, in non-embodied domains such as large language models (LLMs) and vision-language models (VLMs), learned rewards (Christiano et al., 2017; Ziegler et al., 2019; Ouyang et al., 2022) have been key for hard-to-verify tasks like open-ended generation, dialogue, summarization, and perceptual reasoning (Yue et al., 2025; Stiennon et al., 2020; Li et al., 2024).

In the past two years, *embodied* domains such as autonomous driving and household robotics have started developing foundation models with architectures and training paradigms that mirror LLMs and VLMs; a prime example are vision–language–action models (VLAs) (Black et al., 2024; Intelligence et al., 2025; Wang et al., 2025). Similar to the non-embodied tasks described above, in robotics the quality of a behavior is often subjective, hard-to-specify, or defined by fine-grained physical consequences (e.g., tearing part of a napkin while folding it). Thus, there is a growing demand for general-purpose reward models that can evaluate a physical agent's behavior directly from visual observations and a textual task specification (Tan et al., 2025; Lee et al., 2026), reducing the need for costly human evaluation (Atreya et al., 2025) while providing signals for post-training (Amin et al., 2025) and test-time compute (Wu et al., 2025; Kwok et al., 2025).

However, today's embodied reward models fall short of evaluating the nuances of physical behavior in the way humans do. Using real robot videos and human evaluator labels from the RoboArena benchmark (Atreya et al., 2025), we experimentally show that three state-of-the-art (SOTA) embodied reward models systematically *over-reward* behaviors that human evaluators would reject, including low-quality execution, unsafe interactions, and "shortcut" strategies that mimic completion while violating human preferences. The gap between the reward models and human evaluators also widens as the physical task complexity increases from simple pick-and-place to complex bi-manual tool use tasks.

We argue that today's embodied reward models are overly-optimistic because we lack "bad" embodied behavior data necessary for calibration. In non-embodied domains, LLMs and VLMs benefit from exceptionally diverse and scalable reward-training ecosystems in which "bad" data (from toxic language to logical fallacies) is abundant and cheap to generate, enabling a sufficiently large coverage of outcomes and their (human) evaluations. However, embodied data collection is bounded by wall-clock time and hardware safety, making failures, and dangerous behaviors undesirable to

---

[1]UC Berkeley [2]NVIDIA Research [3]Carnegie Mellon University. Email: rantian@berkeley.edu, {yilinwu,abajcsy}@andrew.cmu.edu. This work was partially supported by the National Science Foundation (NSF) CAREER award #2441014. Correspondence to: Thomas Tian <rantian@berkeley.edu>.

collect at scale. Even when failures do occur, they are rarely preserved in the public data ecosystem, as the dominant imitation-based learning paradigm prioritizes expert demonstrations and intentionally filters out negative data (O'Neill et al., 2024). Thus, current embodied AI datasets used for reward model training are biased toward successful behaviors, resulting in models that are systematically over-optimistic.

**Our position is that if we want to have good embodied reward models, we need to invest in more "bad" robot data. We need a deliberate shift away from expert-only datasets and toward the intentional collection and release of dangerous, failed, noisy, and error-prone robot data to obtain reliable embodied reward models.** We demonstrate that even prompting SOTA embodied reward models with real robot failure videos (rather than just textual descriptions of "bad" behavior) as in-context examples results in better preservation of human preferences and fewer high-impact false positives. We call on the robotics and embodied AI community to release curated, large-scale robot failure datasets, develop new methods for the synthetic generation of "bad" robot data, develop more decentralized, physical evaluation systems, and develop benchmarks for evaluating general-purpose embodied reward models.

## 2. Setup: Embodied Reward Models

An embodied reward model assigns (a sequence of) scalar scores to robot behavior conditioned on some task context. Formally, let the task context $c \in \mathcal{V}$ be a free-form language instruction. Let a $T$-length robot behavior be represented by a sequence of high-dimensional observations, $\tau := o_{1:T}$, sensed by the robot's intrinsic and extrinsic sensors (also referred to as a *rollout*). For example, in tabletop robotic manipulation, each $o_t$ typically consists of RGB images collected by a third-person and wrist-mounted camera.

We define an **embodied reward model** as a parameterized map from observations and context to rewards: $R_\theta(o_{1:T}; c) \rightarrow r_{1:T}$, where $r_t \in \mathbb{R}, \forall t \in [0, T]$ is a real value measure of how well the behavior aligns with the user's original task instruction $c$. For example, this can include task progress, execution quality, and safety. The learnable parameters of the model are denoted by $\theta$.

Let an **embodied foundation model** be defined as a parameterized map from an observation and task context to a distribution over actions, $\pi_\phi(\mathbf{a} \mid o, c)$, where $\mathbf{a} := a_{t:t+H}$ is an $H$-step future sequence of actions starting from any timestep $t$, and $\phi$ are the parameters. After an embodied foundation model is pre-trained via imitation on a large dataset of observation-action labels (O'Neill et al., 2024; Khazatsky et al., 2024), the reward function plays a role during post-training and during deployment-time.

During *post-training*, the reward function enables

reinforcement-learning (RL) to improve the precision of the foundation model or to learn new skills (Tian et al., 2024; Zhang et al., 2024; Ghasemipour et al., 2025; Zhang et al., 2025; Tan et al., 2025; Zhai et al., 2025; Amin et al., 2025; Xiao et al., 2025). Mathematically, given an initial observation $o_0$, this process entails solving the following problem:

$$\phi^* = \arg\max_\phi \mathbb{E}_{o_{1:T} \sim \mathbb{P}(\cdot|o_0, \pi_\phi)} \Big[ R_\theta(o_{1:T}; c) \Big],$$

where $o_{1:T} \sim \mathbb{P}(\cdot \mid o_0, \pi_\phi)$ defines the distribution over future observations (i.e., rollouts) induced by the embodied foundation model $\pi_\phi$ under the current model parameters. The reward model scores how aligned the current rollouts are with the task context, and thus directly determines how the policy $\pi_\phi$ is improved.

During *test-time compute*, the embodied foundation model is frozen and samples $\mathbf{a} \sim \pi_\phi(\cdot \mid o; c)$ are taken from the model; let a set of $K$ action samples be denoted by $\{\mathbf{a}^i\}_{i=1}^K$. The reward model scores each action sample and executes only the actions that maximally align with the task (Gao et al., 2024; Wu et al., 2025; Dai et al., 2025):

$$\mathbf{a}^* = \arg\max_{\{\mathbf{a}^i\}_{i=1}^K} \mathbb{E}_{o_{t:t+H} \sim \mathbb{P}(\cdot|o_t, \mathbf{a}^i)} \Big[ R_\theta(o_{t:t+H}, c) \Big],$$

In both regimes, the reward model's calibration establishes an upper-bound for how much the foundation model can be improved, or how reliably it can be steered at deployment-time. As we show in the next section, despite current embodied reward models being able to evaluate temporal progress in the robotics tasks, they systematically over-reward behaviors that humans would reject, such as low-quality execution, unsafe interactions, or "shortcut" strategies that mimic completion while violating implicit preferences.

## 3. Current Embodied Reward Models Fail as Task Complexity Increases

We evaluate three of the latest general-purpose and open-sourced embodied reward models that represent the main families of models used in robotics.

**(1) Preference-based reward models trained with *synthetic* negatives.** These methods construct pseudo-negative or failure-like rollouts from logged expert trajectories, e.g., by shuffling or perturbing successful executions. Preference-learning objectives are used to train the model from the induced contrasts, extracting more supervision per costly interaction. In this work, we focus on the **ReWind** (Zhang et al., 2025) model which is trained via: $\theta^* = \arg\min_{R_\theta} \mathbb{E}_{c, \tau \sim \mathcal{D}} \Big[ \sum_{t=1}^T (r_t - \frac{t}{T})^2 + (r_t^-)^2 \Big]$, where $r_{1:T} = R_\theta(o_{1:T}; c)$ are the model predictions for a given ob-

servation sequence, and $r_{1:T}^- = R_\theta(o_{1:T}^-; c)$ are reward predictions for a *synthetic negative* observations sequence, $o_{1:T}^-$. We follow the open-sourced code from (Zhang et al., 2025) and train **ReWind** on the large-scale Open-X embodiment dataset (O'Neill et al., 2024) of expert robot demonstrations.

**(2) Zero-shot VLM rewards.** An increasingly popular alternative is to cast reward prediction as an embodied reasoning task. These approaches leverage VLMs as zero-shot reward predictors by first asking the model to articulate its evaluative reasoning (e.g., perform a frame-level analysis of $o_{1:T}$), and then generate a dense reward sequence grounded in that reasoning. The approach we follow in this work is called **GVL** (Ma et al., 2024), and we query GPT-5 (OpenAI, 2025) as the VLM. We show the detailed prompt in the Appendix A.

**(3) Fine-tuned VLM rewards.** Finally, VLMs have also been fine-tuned to be reward predictors, aiming to transfer the VLM's semantic priors into task-specific embodied scoring functions. Here, fine-tuning signal typically comes from automatically or heuristically constructed task progress labels (Amin et al., 2025; Lee et al., 2026). The pretrained, open-sourced model we use in this work is called **Dopamine** (Tan et al., 2025) which fine-tunes a pre-trained VLM through regression to predict rewards (Appendix A).

### 3.1. Evaluation Protocol

Our evaluation dataset $\mathcal{D}_{\text{eval}}$ consists of a small number of real robot rollouts drawn from the same task description $c$ and (similar) initial conditions: $(\{o_{1:T}^i\}_{i=1}^N, c) \in \mathcal{D}_{\text{eval}}$. We aim to quantify the "goodness" of the predicted rewards $r_{1:T}$ from any of the above reward models. One of the key challenges for doing reward evaluation is that we do not have access to dense oracle rewards. Prior works often use the Value-Order Correlation (VOC) metric (Ma et al., 2024; Tan et al., 2025; OpenGVL Team, 2025). It measures the rank correlation (e.g., Spearman) between the predicted values and the ground-truth temporal order of frames in expert videos. Thus, larger VOC scores indicate better temporal progress understanding. However, this metric often misses fine-grained aspects of *how* a task is being executed. For example, consider the behavior shown on the left-hand side of Fig. 2. This robot behavior exhibits coarse temporal progress (e.g., moving an object from one position to another) but it causes a safety violation along the way (e.g., toppling the blue bowl).

Therefore, in our evaluation, we treat human judgments—specifically, pairwise human comparisons of two rollouts sampled from the dataset $o_{1:T}^A, o_{1:T}^B \sim \{o_{1:T}^i\}_{i=1}^N$—as the ground-truth signal for evaluation. For each task, we aggregate human pairwise comparisons on the robot's behavior into an ordinal preference ranking over trajectories. We

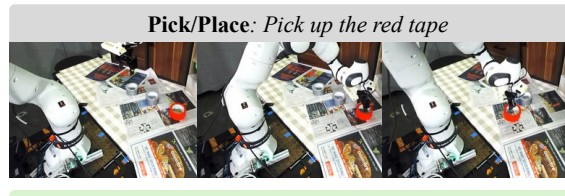

**Pick/Place**: *Pick up the red tape*

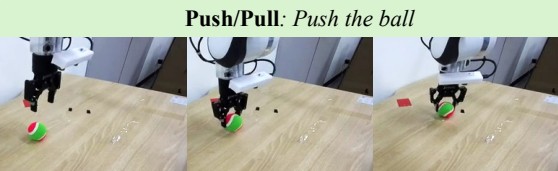

**Push/Pull**: *Push the ball*

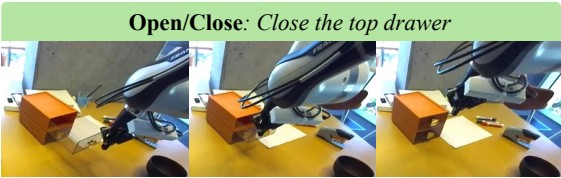

**Open/Close**: *Close the top drawer*

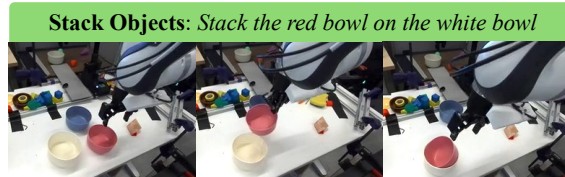

**Stack Objects**: *Stack the red bowl on the white bowl*

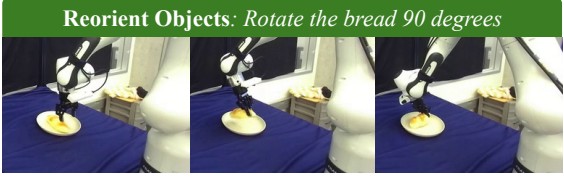

**Reorient Objects**: *Rotate the bread 90 degrees*

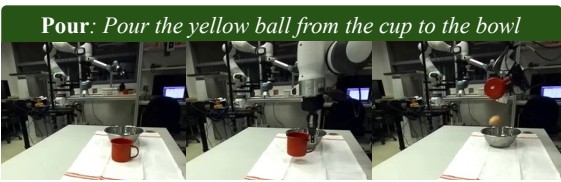

**Pour**: *Pour the yellow ball from the cup to the bowl*

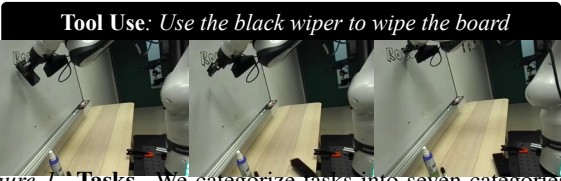

**Tool Use**: *Use the black wiper to wipe the board*

*Figure 1.* **Tasks.** We categorize tasks into seven categories of increasing complexity: Pick/Place, Push/Pull, Open/Close, Stack Objects, Reorient Objects, Pour Liquid, Tool Use. The videos above are from RobotArena (Atreya et al., 2025).

then compute a reward model's return for each rollout by accumulating per-step rewards, inducing a corresponding ranking. Each reward model is evaluated by how well its induced ranking agrees with the human ranking. We describe the procedure and metrics below in detail.

**Robot Behavior & Human Evaluation Dataset.** We use the real robot behaviors and human evaluations from RoboArena (Atreya et al., 2025), a decentralized evaluation

framework for generalist robot policies. Tasks are predominantly performed by robotic manipulators, but deployment conditions and tasks are varied from simple picking tasks (e.g., *"Pick up the red paper"*) to more precise motions (e.g., *"Pour the nuts from the red cup onto the plate"*). Specifically, we study seven task categories of increasing complexity (visualized in Fig 1): **Pick/Place** objects on a tabletop, **Push/Pull** interactions with rigid objects, **Open/Close** interactions with articulated objects (e.g., door), **Stack Objects** on top of each other, **Reorient** objects to new poses, **Pour** items from one container into another, and **Tool Use** (e.g., picking up a whiteboard marker and erasing a whiteboard). Our tasks span both rigid-body, soft-object manipulation (e.g., cloth folding), as well as multi-step tasks (e.g., multi-stage tool use).

In each RoboArena evaluation trial, an evaluator selects a task as a natural-language instruction that describes one of these tasks, and the benchmark queries $n$ policies $\{\pi^1(\mathbf{a} \mid o, c), \ldots, \pi^n(\mathbf{a} \mid o, c)\}$ (hidden, pre-trained, and randomly selected) each of which is executed on the same robot with similar initial conditions. The full rollout consisting of both observations and actions are logged for the human evaluator to inspect. The human evaluator assigns each rollout a single continuous performance score (e.g., 0 for failure, 100 for success, and intermediate values for partial success). In addition, the evaluator provides an explicit preference judgment (i.e., which rollout is better) along with a short rationale in text describing why (e.g., stability of the grasp, placement quality, safety-relevant contact, or shortcut behaviors) on a representative A/B comparison (two rollouts from the same task context that illustrate a meaningful qualitative difference).

In our work, for any tuple $(\{o_{1:T}^i\}_{i=1}^n, c) \in \mathcal{D}_{\text{eval}}$, let the corresponding set of scores given by a human evaluator be denoted by $\{y^i\}_{i=1}^n$ where $y^i \in [0, 100]$. We use these human-annotated rollout scores to induce a ground-truth ordering over rollouts within each task context, and curate an evaluation set that spans the seven task categories of increasing complexity and nuance described earlier.

**Evaluation Metrics.** Given a reward model $R_\theta$, we compute the model's score $\hat{y}^i$ for each rollout by accumulating the predicted per-step rewards: $\hat{y}^i = \sum_t \hat{r}_t$, where $\hat{r}_{1:T} = R_\theta(o_{1:T}; c)$. We first measure the human-model disagreement by looking at the pairwise ranking mismatch. Specifically, for a task context $c$, let the set of rollout pairs with a strict human preference be: $P_c = \{(i, j) : i < j, \ y^i \neq y^j\}$. For any such pair $(i, j) \in P_c$, let the *sign* of the human's preference be denoted by $s_{\text{H}}^{ij} = \text{sign}(y^i - y^j)$ and the sign of the reward model's preference be denoted by $s_{\text{M}}^{ij} = \text{sign}(\hat{y}^i - \hat{y}^j)$. Finally, the per-task **human-model disagreement**: $D_c = \frac{1}{|P_c|} \sum_{(i,j) \in P_c} \mathbf{1}\left[s_{\text{H}}^{ij} \neq s_{\text{M}}^{ij}\right]$, which

is minimized ($D_c = 0$) when the reward model agrees with the human preference direction on every strictly preferred pair in $P_c$, and maximized ($D_c = 1$) when it disagrees on all such pairs. We measure the overall **preference ordering accuracy** by aggregating the disagreement across tasks, weighted by the count of pairs in $P_c$: $A = 1 - \frac{\sum_c |P_c| D_c}{\sum_c |P_c|}$. Intuitively, $A$ is maximized ($A = 1$) when the reward-induced ordering matches the human ordering for every comparable pair, and decreases toward 0 as the model increasingly reverses human preferences.

### 3.2. Results

**Quantitative Results.** Fig. 3 visualizes the preference ordering accuracy of **ReWind**, **GVL**, and **Dopamine** across increasingly complex and nuanced-to-evaluate tasks. On visually simplistic goal-reaching tasks such as **Pick/Place**, all models achieve relatively high preference-ordering accuracy (0.72–0.77), indicating they can often rank clearly different outcomes correctly. However, performance drops steadily as tasks require finer execution quality, temporal coordination, or implicit constraints. For categories like **Reorient Object** and **Pour**, accuracy falls into the low-to-mid 0.6 range, and for the most nuanced task, **Tool Use**, performance is only modestly above random guessing (0.52–0.62).

**Qualitative Results.** To better understand why the reward models degrade, consider the example (*not* from RoboArena, but collected on our own hardware) shown on the left of Fig. 2 for the task context $c$=*"Use two hands to lift the lid and carefully put it on the table without colliding with the bowl."* The robot video appears to make progress along the task (hands reach the lid, the lid is lifted, and lid approaches the table), yet the execution violates the instruction to *not* collide with the bowl: the lid knocks into the bowl. Nevertheless, all reward models predict high reward values. In other words, the reward model over-weights progress-like signals that only *correlate* with success in its training distribution, but it under-penalizes the negative behavior that humans use to distinguish acceptable from unacceptable execution. A similar example is shown on the right side of Fig. 2 for the task context $c$=*"Pour the nuts from the red cup onto the plate."*. The robot rollout appears to make steady progress yet nuts spill outside the plate. Despite this failure (shaded region), both **GVL** and **ReWind** continue to assign increasing per-frame rewards.

**Analysis.** A high-quality reward function ultimately requires understanding the contrast between "good" and "bad" behavior. All families of reward models described in Sec. 3 heavily rely on non-embodied priors (e.g., from large-scale VLM pre-training) or heuristics (e.g., hand-engineered criteria, synthetic counterexamples) as *substitutes* for the strong learning signal that would come from explicit negative em-

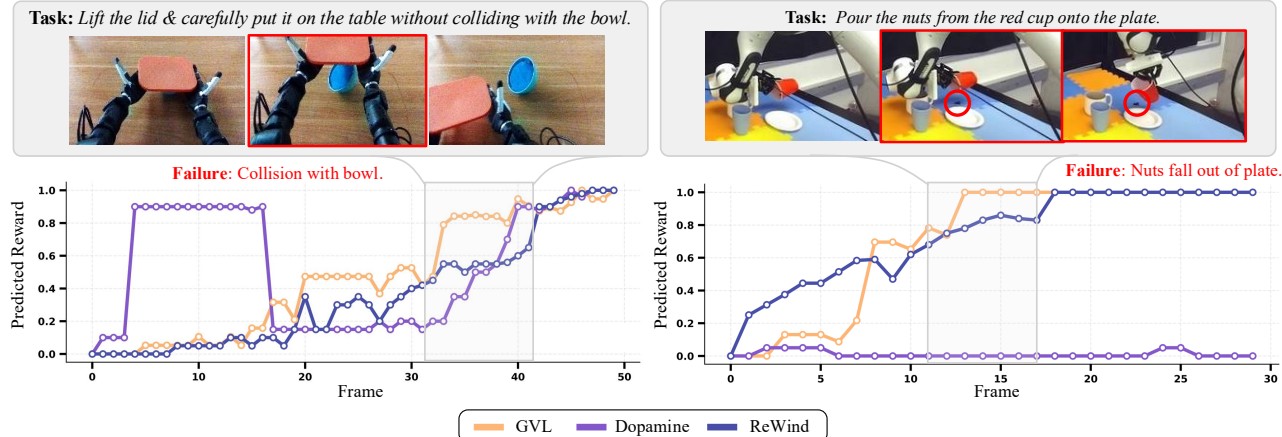

Figure 2. **Dense Reward Predictions from Embodied Reward Models.** We evaluate three SOTA embodied reward models on real robot failure videos. *Top*: Representative failure frames. Red circles highlight the human-identified failure event (e.g., collision with the bowl or the nuts spilled outside of the plate). *Bottom*: Per-frame predicted reward. The shaded interval indicates the visualized negative-behavior segment. All reward models assign increasing reward values during the video, even during the failure frames.

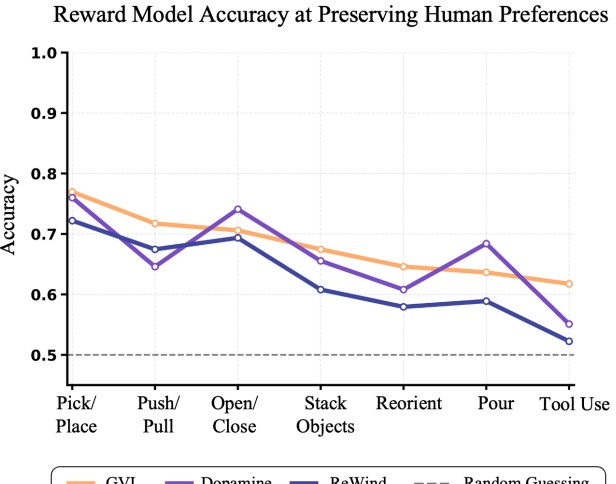

Figure 3. **Preference Ordering Accuracy as a Function of Task**. We measure how often each reward model's induced returns recover the human ordering over rollouts (fraction of human-preferred pairwise comparisons correctly ranked). Tasks are arranged in increasing complexity and the dashed line indicates random guessing (0.5). Reward models (GVL, Dopamine, ReWind) approach random guessing as the task complexity increases.

bodied failure data. For example, **ReWind** constructs negatives by perturbing successful rollouts, but such negative examples cannot faithfully represent closed-loop failure modes that dominate real deployments. **GVL** inherits generic scoring criteria from VLM pretraining rather than user calibration; consequently, it is poorly calibrated on safety- and quality-critical failures that are underrepresented in the pretraining distribution. **Dopamine** relies on heuristically constructed supervision (progress- or end-state surrogates), which is biased to favors behaviors that are "getting there" but overlook the fine-grained quality of the execution or the safety properties of the behavior.

**Implications.** As discussed in Section 2, the reward model is the optimization objective in both RL post-training and test-time compute (and, of course, autonomous evaluation). Thus, reward model calibration sets an upper-bound on performance and on evaluation. As embodied foundation models begin generating increasingly complex behaviors, our results suggest a widening gap between model capability and the capacity of current reward model paradigms to evaluate them.

# 4. Injecting "Bad" Robot Data into Embodied Reward Models Improves Performance

Our position is that if we want good embodied reward models—especially ones that can evaluate complex robot behaviors—we need to invest in more "bad" robot data. To substantiate this claim, in this section, we demonstrate that even modest amounts of "bad" robot data can improve the calibration of off-the-shelf reward models and boost their alignment with real human preferences. We focus on **GVL** (Ma et al., 2024) because this model enables us to inject negative examples via in-context learning, without any re-training. We control *how* the negative data is fed to the model: from easy-to-generate but lossy *textual* descriptions of robot failures, to *videos of real robot failures*, to also prompting with *dense reward labels*.

**Text Descriptions of Failure.** For each task in our evaluation dataset, we retrieve the corresponding free-form text feedback provided by the RoboArena evaluators. We use an LLM to distill the feedback into a concise but general summary of common failure behaviors for that task (e.g., *"the robot grasps the correct object but fails to release"*, *"approaches the goal but contacts a forbidden region," "moves toward completion while leaving the object unstable"*). We prompt **GVL** with these failure descriptions as explicit neg-

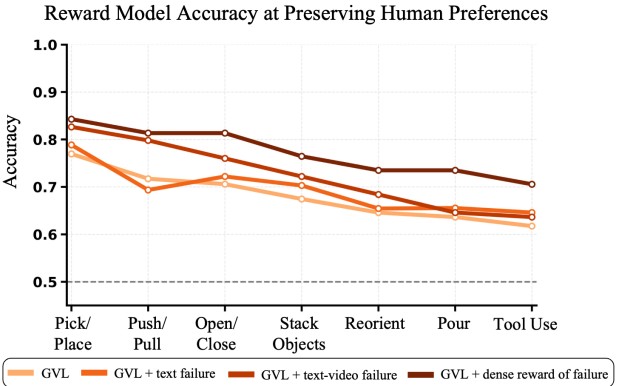

Reward Model Accuracy at Preserving Human Preferences

*Figure 4.* **Effect of In-Context Negative Robot Data on Reward Accuracy.** We use the GVL model throughout (Ma et al., 2024). Text-only negatives provide little to no improvement, while grounding negatives in the actual failure visuals improves ordering on simpler categories. The largest gains come from dense reward examples of negative behavior, especially in more nuanced categories where "bad" behavior is defined by execution quality and safety constraints rather than coarse task progress.

ative examples, before asking the model to score a new behavior.

**Text Descriptions & Robot Videos of Failures.** Building on the text-only setting, we additionally pair each failure description with a real robot video which exhibits this failure mode. Concretely, for each task, we pair the distilled failure cues with sampled observation snippets. These text–image pairs are provided as in-context examples to the GVL model. Our hypothesis is that videos provide finer-grained features for evaluating whether a behavior is good or bad; features about time-varying behavior that are difficult to capture using natural language alone.

**Text Descriptions, Robot Videos, and Dense Reward Labels of Failures.** Finally, in addition to text descriptions and videos, we also provide dense per-timestep rewards associated with a negative behavior example. Since oracle dense rewards are unavailable (recall, RoboArena only provides a single scalar value for the *entire* robot behavior), we perform preference-guided self-distillation to obtain proxy dense rewards. Intuitively, we use sparse human preferences from RoboArena to select between different dense reward labels produced by the reward model, and return a dense reward that is maximally aligned with the real human preferences. Specifically, for each A/B test rollout pair in RoboArena's evaluation task, we query the GVL model multiple times (10 samples in our implementation with a temperature of 0.8) to obtain a set of dense reward sequences, $r_{1:T}$, for the same pair of rollouts. Each candidate reward sequence induces an overall return by summing rewards over time, and thus an implied ordering between A and B. We retain the reward sequence whose induced ordering matches the human preference from RoboArena, and use the reward sequence of the *less-preferred* robot behavior as an in-context example of how negative behavior should be evaluated.

**Results.** In Fig. 4, we show how different in-context negative examples affect GVL's ability to preserve human preference orderings. Although text descriptions of failures are easy to generate (and are conceptually related to Constitutional AI approaches to injecting understanding of harmful consequences into foundation models (Bai et al., 2022; Sermanet et al., 2025)), they do not significantly improve GVL's evaluation quality. This suggests that despite the powerful LLM backbone, current SOTA VLMs still struggle to ground textual criteria into concrete physical, embodiment, and viewpoint-specific robot behaviors.

Prompting GVL with text and robot failure videos results in a $\approx 8\%$ meaningful accuracy improvement on simple tasks like **Pick/Place** and **Push/Pull**. When inspecting these tasks, we see that failures tend to be visually "obvious": grabbing the wrong object, completely missing a grasp, or failing to place an object in the correct location. Grounded counterexamples help disambiguate what "bad" behavior looks like under the task context. As illustrated in the left example of Fig. 5, providing text along with a representative failure rollout enables GVL to sharply down-weight the corresponding behavior, assigning near-zero rewards after failure occurs where the base model would otherwise remain overly optimistic.

However, as the task complexity increases towards **Tool Use**, the goodness of behavior becomes more subtle: for example, the robot's behaviors appear to make progress, but violate implicit quality or safety criteria, like selecting *unstable* places to put objects, making *unsafe* contact with the environment, or short-cutting execution. Here, the negative text and video examples are no longer enough to improve the performance of the reward model. However, this is exactly where dense negative reward examples are most effective: by providing a temporally resolved penalty trace for a human-rejected rollout, the reward model receives a concrete calibration signal for how these nuanced constraints should be reflected in reward space, rather than relying on coarse progress cues or a small number of visually grounded negatives. The right example in Fig. 5 demonstrates this effect. The rollout initially looks plausible and "on track," but the robot drops the wiper within the highlighted failure window; only when provided with a dense negative reward example does GVL sharply down-weight the trajectory at the failure moment. Overall, negative reward examples result in an approximate $10\%$ accuracy improvement even in the hardest **Tool Use** and **Pour Liquid** tasks. While our analysis focuses on tabletop manipulation, the same principle naturally extends to other embodied domains such as navigation and long-horizon tasks, which are important extensions for future work.

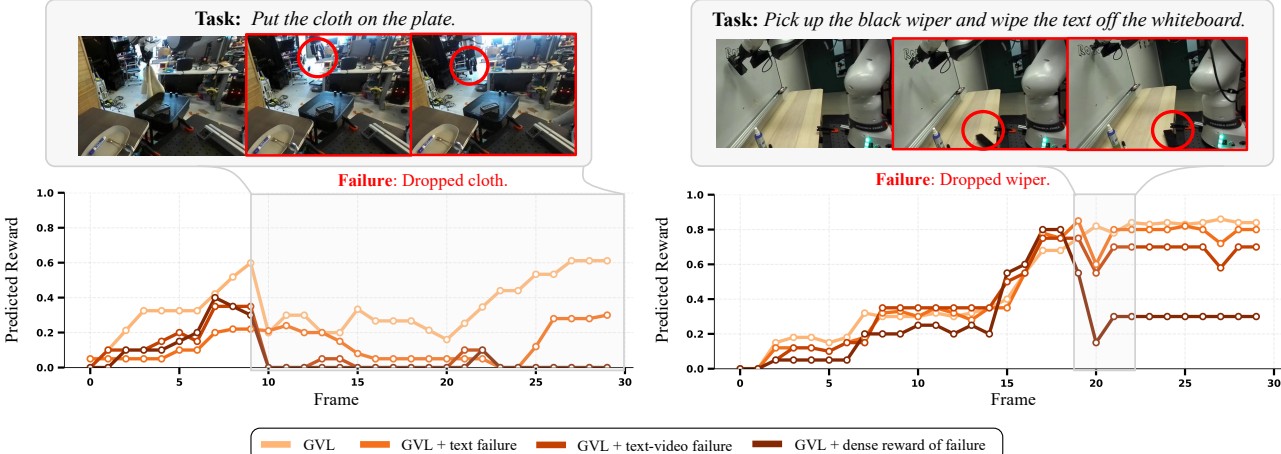

*Figure 5.* **Influence of "Bad" Robot Data on Reward Predictions.** We prompt the **GVL** reward model with three levels of in-context negative data: text-only, text-video pairs, and dense reward labels of robot failure videos. Videos of real robot failures along with dense reward labels helps the model catch subtle errors, like the robot dropping the black wiper at frame 20 on the right-hand video.

## 5. Open Problem: Embodied Preference Datasets with Diverse Failures

Our experiments in Section 3 showed that embodied reward models become more calibrated when they see real failure robot data *and* when they are informed of how "bad" a behavior appears from human feedback. It stands to reason then, that we need to contend with two problems if we want good embodied reward models: (1) where do we get more "bad" robot data from? and (2) how do we get human feedback on this data?

In theory, one might hope to scale embodied data and human feedback collection by borrowing from the LLM playbook: run many replicas of a foundation model in parallel, have each instance generate diverse rollouts (mixture of good and bad), upload logs to a central database, and then perform large-scale offline evaluation in parallel. This strategy works exceptionally well in non-embodied settings because the entire loop is highly parallelizable: 1) sampling model rollouts is a purely digital workload that scales with access to more compute, 2) the foundation model *directly generates* human-consumable outputs (i.e., text, images) that are immediately "judgeable", and 3) preference labeling can be distributed across annotators who can directly compare logged outputs with high throughput. Thus, negative data generation has a primarily digital (rather than physical) impact, and the model generations directly enable scalable human feedback.

However, in embodied settings like robotics, producing the rollouts is the dominant cost. First, the embodied foundation model generates *actions*, $a_{t:t+H}$, which are not easy to evaluate directly; furthermore, actions and outcomes are not one-to-one mapping due to stochasticity of the world). Thus, the embodied foundation model's generations need to be translated to *physical outcomes* for evaluation (i.e., the observations $\{o_{1:T}\}$). Today, there still exists no univer-

sally good substitute for this translation process except for *actually executing the generated actions on robot hardware*.

## 6. Alternative Views

**Pretrained VLM reward models have already seen bad-behavior data.** Today's VLMs are pretrained on web-scale data containing failures, accidents, and unsafe interactions, so one might argue VLM-based reward models already inherit sufficient exposure to bad behavior. Our claim is *not* that VLM-based reward models have never seen failure data; rather, generic pretraining remains an *imperfect substitute* for the specific kind of negative supervision needed in embodied reward modeling—examples that are *embodied*, *human-calibrated*, and directly informative about which robot behaviors should be penalized. Consequently, VLM-based reward models may still be overly optimistic on subtle failure cases humans would reject.

**The core problem isn't data balance; its observability.** Perhaps the reason why our embodied reward models are failing to evaluate behaviors as accurately as humans do is because they don't get access to the same rich, low-level embodied signals: forces, tactile signals, sound, temperature, smell, and taste. In other words, our current embodied reward models don't have sufficiently rich representations of the world, and so the key variables needed for evaluations are unobservable. Limited observability is undoubtedly a challenge, and today's reward models have only scratched the surface of all modalities of embodied data. However, observability issues cannot fully explain the systematic over-rewarding we observe in our experiments, since humans are able to accurately evaluate these behaviors even without access to low-level embodied signals like forces, tactile, etc.

**"Bad data" is neither scalable nor necessary; uncertainty**

**quantification of the reward model is.** Obtaining good coverage of robot failures via the deliberate collection of "bad" data is fundamentally limited, especially in safety-critical and human-centered environments. Instead of investing in this costly data collection, an alternative approach is to leverage uncertainty quantification (UQ) (Kuleshov et al., 2018; Angelopoulos et al., 2023) to achieve more calibrated reward models. Prior works have demonstrated that UQ methods can improve the robustness, reduce over-confidence, and improve alignment of reward models in the large language domain (Leng et al., 2025; Park et al., 2025). However, with access only to positive data, it is extremely challenging to calibrate the false positive rate of a reward model, as false positives are defined with respect to unobserved negative outcomes.

**We should use verifiable rewards, not learned ones.** Perhaps learned reward models are the wrong target, as they require strong coverage of outcomes that are hard to get in embodied settings. Instead, we should design *verifiable rewards*: for example, using Signal Temporal Logic (STL) to specify the reward criteria (Kress-Gazit et al., 2024), can allow the specification of fine-grained criteria, enable easier debugging, and alleviate the data burden because of the priors encoded within the reward specification language. At the same time, verifiable rewards have been historically difficult to scale to the full complexity of the open-world; however, emerging research indicates that these specification strategies may be compatible with the latent states inside of pre-trained models (Kapoor et al., 2025), pointing to future work where verifiable and learned rewards are complementary.

## 7. Call to Action

We analyzed today's embodied reward models and found they are systematically over-optimistic when compared to human evaluators. We argue that reliable embodied reward models require a deliberate shift beyond expert-only datasets toward the intentional collection and release of dangerous, failed, noisy, and error-prone robot data. Our calls to action are as follows.

**Action #1: Release curated, large-scale robot failure datasets**. We call on industry teams operating robot fleets, academic labs running real-robot testbeds, and benchmark organizers to establish consistent pipelines for releasing negative robot behavior data. In particular, we advocate for the creation of failure-focused datasets at a scale comparable to today's large expert demonstrations (O'Neill et al., 2024; Khazatsky et al., 2024). At present, most public robot datasets (and the benchmarks built on top of them) contain almost exclusively successful behaviors, leaving reward models with little to no exposure to failures, unsafe interactions, or degraded execution (OpenGVL Team, 2025). Similar challenges have existed in adjacent domains, such

as autonomous driving, where safety-critical failure data is largely proprietary and only sparsely released (e.g., Nexar crash dataset (Moura et al., 2025)). From our experiments, informative failures are those that are safety-critical or otherwise strongly misaligned with user preferences, especially when behavior may still appear to make progress or even resemble task completion on the surface (Sec. 3); more broadly, we also hypothesize that failures arising from long-horizon causality, failures that require temporal information to evaluate correctly, and failures on which current models exhibit high uncertainty may be important for reward modeling. Our goal is not necessarily to induce dangerous behavior (although this is plausible in controlled lab or field tests), but rather to stop discarding negative data that already arises during routine operation, debugging, and deployment. Standardizing how failures are captured, labeled, and shared (through common data formats, failure taxonomies, and privacy-conscious release practices) would transform these currently siloed datasets into reusable calibration for embodied reward models.

**Action #2: New methods for the synthetic generation of "bad" robot data.** We call for increased investment in methods that enable the *synthetic generation* of failed and unsafe robot behaviors. The community should prioritize high-fidelity simulators (Makoviychuk et al., 2021; Todorov et al., 2012; NVIDIA et al., 2025) that support physically interactive, contact-rich, and deformable tasks, where real-world failure data is hardest to collect. In parallel, real-to-sim approaches (Jangir et al., 2025) should advance beyond static or quasi-static reconstructions toward physically interactive real-to-sim models that allow counterfactual perturbations of real trajectories, while maintaining high-fidelity geometry and visual realism (e.g., Gaussian splats, NeRFs). Advances in image and video editing foundation models provide a complementary path for synthetic failure generation. For example, the ASIMOV dataset (Sermanet et al., 2025) demonstrates how unsafe embodied images can be synthesized from safe ones; however, current approaches are largely limited to static imagery. To fully support robotics applications, these methods must be extended to generate temporally consistent, interactive failure trajectories. Here we identify an exciting opportunity to extend current action-conditioned (video) world models (Agarwal et al., 2025; Hafner et al., 2025; Zhou et al., 2024; Guo et al., 2025; Mei et al., 2026; Assran et al., 2025): to generate not only successful outcomes but also imagine (diverse compositions of) failed outcomes from small amounts of real failure data. Importantly, these synthetic generators should be understood as *complementing* rather than replacing real bad-behavior data: simulation alone struggles with sim-to-real gaps, complex dynamics, and rare edge cases, while the most powerful regime is a *real2sim2real* loop in which deployed failures refine simulators, simulators produce richer synthetic fail-

ures, and these in turn calibrate the reward model before further deployment exposes new edge cases.

**Action #3: More decentralized, physical evaluation systems.** The RoboArena (Atreya et al., 2025) benchmark utilized in this work is a first-of-its-kind multi-site evaluation system with real human evaluations, running multiple generalist policies, all deployed in diverse real-world environments. It also shows that decentralized evaluation is feasible and substantially improves ecological validity and coverage. At the same time, RoboArena makes the practical scaling constraints concrete: matching the initial conditions between trials, ensuring safety of *any* policy executed on hardware, and performing resets imposes operational friction that caps how quickly robot data and human feedback can be obtained. Scaling this paradigm is therefore not just a matter of "adding more sites," but of establishing shared infrastructure—standardized rollout procedures (initial-state matching, safety gating, and reset protocols), logging formats, evaluation rubrics, and lightweight quality control—so that results are comparable across sites without incurring prohibitive per-run overhead.

**Action #4: New benchmarks for evaluating general-purpose embodied reward models.** Finally, we call on the community to invest in better ways to evaluate general-purpose embodied reward models themselves. Existing evaluations remain limited in both scale and supervision quality. Even in the most recent efforts (Lee et al., 2026), labels are often sparse and are frequently derived from heuristics or VLM-based surrogates rather than real human feedback. We believe that, while such proxies can provide coarse assessment of if a reward model is on the right track or not, evaluation labels should be grounded in what matters to real human stakeholders. If evaluation is not grounded in real human stakeholder values, then models can appear accurate under proxy scores while failing precisely in the cases that matter for downstream optimization and deployment. We therefore call for the development of reward model benchmarks that provide dense reward labels on real robot data, annotated by high-quality human evaluators.

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

# A. Embodied Reward Models

## A.1. Prompt used for GVL

GVL (Ma et al., 2024) is a generative reward model that frames reward prediction as an embodied reasoning task. Concretely, it uses a vision-language model (VLM) as a zero-shot reward predictor by first eliciting explicit evaluative reasoning over the rollout (e.g., a frame- or segment-level analysis of observations $o_{1:T}$ conditioned on the task instruction), and then producing a dense per-step reward sequence grounded in that reasoning. The original GVL implementation uses Gemini as the default VLM backbone; in our experiments, we instead use GPT-5 due to stronger empirical performance. Unless otherwise noted, all GVL scores are obtained from a single model query per trial with a temperature of 0.6. In the following, we show the detailed prompt used in different experiments. The prompt supports three optional conditioning modules: (i) in-context reference behaviors with known rewards, (ii) task-specific text-only failure descriptions, and (iii) visual failure references (failure-example frames).

---

**GVL Reward Prediction Prompt (template)**

**System:** You are an expert roboticist and safety-aware evaluator.

**User:** Predict task reward for frames of a robot performing the task:

$$\texttt{TASK} = \{\texttt{TASK}\}$$

The task reward is between 0 and 100, where 100 corresponds to successful task completion. The evaluation should consider both the robot's behavior and the user's preferences, including undesired or unsafe behavior. If the robot approaches undesired actions, the reward should be low; if it clearly violates safety constraints, the reward can be 0. **Frames are presented in random order; evaluate each frame independently.**

**Input: Initial scene (anchor frame).**
*[ANCHOR_IMAGE]*
Constraint: the reward for the initial frame must be 0.

**Optional module A: In-context reference behaviors (with known rewards).**
Each reference behavior is a set of frames shown in random order:
• **Reference behavior {d}**, frame {j}, known reward: {r}. *[DEMO_IMAGE]*

**Optional module B: Task-specific failure descriptions (negative cues).**
Use only if the current frame *visibly matches*; do *not* assume failures occurred. If a frame clearly matches a listed failure behavior, lower the reward accordingly.
  1. {failure_desc_1}
  2. {failure_desc_2}
  3. {failure_desc_3}

**Optional module C: Visual failure references (negative examples).**
Use only as visual references; do *not* assume they occurred in the current rollout.
• **Failure example {e}**, frame {j}. *[FAILURE_EXAMPLE_IMAGE]*

**Task:** For the following rollout frames shown in random order, output a JSON array:

```
[
  {"frame_number": i,
   "frame_description": "...",
   "task_reward": 0-100}
]
```

**Guidelines:**
• Describe only what is visible in the frame.

• Use negative cues only as evidence; do not hallucinate failures.

• Do not assume the current frames match the failure examples.

**Frames:**
*Frame 1: [IMAGE_1]*
*Frame 2: [IMAGE_2]*
. . .
*Frame N: [IMAGE_N]*

---

### A.2. Fine-tuned VLM rewards.

We use the open-sourced **Dopamine** reward model in our evaluations. Unless otherwise specified, we use the 8B checkpoint in a zero-shot setting. **Dopamine** takes as input synchronized three-view video observations together with a goal image. For each task, we construct the goal image from the final frame of a held-out successful rollout, and remove that rollout from the evaluation set to avoid evaluating on the same trajectory used to define the goal.

### A.3. Robot Behavior & Human Evaluation Dataset.

We use the real robot behaviors and human evaluations from RoboArena (Atreya et al., 2025), a decentralized evaluation framework for generalist robot policies. Tasks are predominantly performed by robotic manipulators, but deployment conditions and tasks are varied from simple picking tasks (e.g., *"Pick up the red paper"*) to more precise motions (e.g., *"Pour the nuts from the red cup onto the plate"*). We used 723 valid tasks in our evaluation; each task contains from 2 to 6 rollouts. Specifically, we study seven task categories of increasing complexity (visualized in Fig 1): **Pick/Place** objects on a tabletop (427), **Push/Pull** interactions with rigid objects (32), **Open/Close** interactions with articulated objects (e.g., door) (72), **Stack Objects** on top of each other (47), **Reorient** objects to new poses (68), **Pour** items from one container into another (30), and **Tool Use** (e.g., picking up a whiteboard marker and erasing a whiteboard) (47).

