# OpenReview forum: "Position: Good Embodied Reward Models Need Bad Behavior Data"
_ICML.cc/2026/Position_Paper_Track — ICML 2026 Position Paper Track spotlight_

### Official Review · Reviewer_Gbia · 2026-03-12

**Significance:** 3
**Argument Clarity:** 3
**Rating:** 5
**Confidence:** 3

**Questions:**

See above.

**Alternative Views Section:**

Yes

**Compliance With Llm Reviewing Policy A Conservative:**

Affirmed.

**Discussion Potential:**

3

**Final Justification:**

The authors have addressed my questions.

**Paper Summary:**

The paper argues that building good reward models should include bad-behaviour data.
The paper shows empirically that three types of existing reward models over reward behaviours that humans deem unsafe, poor, or unrealistic.
Furthermore, by adding some bad-behaviour data, the resulting reward model can improvement alignment with human preferences.
The paper suggests researchers to release large-scale bad-behaviour datasets, new methods to generating synthetic bad-behaviour data, decentralized physical evaluation, and new benchmarks for evaluating reward models.
Finally, the paper describes three alternative views: data observability, uncertainty quantification, and verifiable rewards.

**Position:**

Yes

**Position In Title:**

Yes

**Related Work:**

3

**Strengths And Weaknesses:**

## Strengths
- The experiments in section 4 provide evidence that GVL could be lacking bad-behaviour data. Notably by injecting text description, video, and dense-reward of failure in the context the reward model improves in performance compared to that of without the injection.
- The paper focuses on four potential directions that researcher can act upon.

## Comments
- While I think the general message is supported by section 4, I have trouble finding section 3 to be supporting. While ReWind is trained purely on expert demonstrations, the two remaining variants leverage a pretrained VLM which might have included some bad-behaviour data. While I agree that the result from ReWind might suggest that missing bad-behaviour data could be the reason why it misaligns against human evaluator, it is not as convincing for the other two models.
  - Secondly, the paper suggests that finer-execution quality, temporal coordination, and implicit constraints might be the cause of performance drop---all of which do not necessarily correlate with bad-behaviour data. Ideally the experiments should decouple these other error modes, for example in temporal-coordination tasks one can break down the tasks into separate subtasks and evaluate whether the reward models are aligned with human evaluators.
- Regarding Section 5, perhaps the paper should address why one cannot use simulation data to generate bad-behaviour---for example in the temporal-coordination tasks with rigid items, a high-fidelity simulation should provide good data to train a reward model.
- Is there evidence that the uncertainty quantification methods are known to have challenges calibrating the false positive rate?

**Support:**

3

---

> ### Author Rebuttal · Authors · 2026-03-31
>
> We would like to thank the reviewer for taking the time to review our paper and provide valuable feedback! We appreciate the constructive feedback that helps us further improve the paper!
>
> ***Q1. While I agree that the result from ReWind might suggest that missing bad-behavior data could be the reason why it misaligns against human evaluators, it is not as convincing for the other two models  (since they might have seen bad data).***
>
> A1. We would like to thank the reviewer for raising this important question. We would like to clarify that our position is not that the poor performance of VLM-based reward models is due to never having seen bad-behavior data at all. Rather, our argument is that generic VLM pretraining is still an imperfect substitute for the specific kind of negative supervision needed in embodied reward modeling, namely examples that are embodied, human-calibrated, and directly informative about which robot behaviors should be penalized. Consequently, VLM-based reward models may still be overly optimistic when evaluating subtle failure cases that humans would reject.
>
> ***Q2. Finer execution quality, temporal coordination, and implicit constraints might be the cause of the performance drop, all of which do not necessarily correlate with bad-behavior data.***
>
> A2. We thank the reviewer for this insightful comment. Our intention is not to claim that missing bad-behavior data is the sole cause of reward model failures. Rather, our claim is that these are the kinds of failures where explicit negative supervision may be especially helpful for improving alignment with human judgment. In such cases, reward models may rely too heavily on coarse progress cues while failing to learn which subtle failures humans would actually penalize. Moreover, Section 4 provides preliminary evidence that more grounded negative data is particularly helpful in these settings. More broadly, there are indeed multiple factors affecting the performance of reward models; our focus in this paper is to highlight missing bad-behavior supervision as one important and currently under-addressed contributor. We will edit the paper to make this scope and argument clearer.
>
>
> ***Q3. Why cannot one use simulation data to generate bad-behaviour?***
>
> A3. We agree that simulation is a promising source of bad-behavior data, and this is exactly in line with our Action #2 on synthetic bad robot data generation. At the same time, simulation-based bad data generation remains an important research frontier, and relying on simulation alone may not be sufficient, since failures often depend on sim-to-real gaps, complex dynamics, and rare edge cases that are difficult to fully capture in simulation. Furthermore, real failure data can itself support a real2sim2real pipeline, where real-world failures help refine simulators, improved simulators generate more informative synthetic failures, and these failures in turn help improve the reward model before further deployment reveals new edge cases. We therefore view simulation as complementary to real-world bad-behavior data, rather than a replacement for it.
>
> ***Q4. Is there evidence that the uncertainty quantification methods are known to have challenges calibrating the false positive rate?***
>
> A4. We would like to clarify that our intention is not to claim that UQ cannot calibrate false positives. Rather, our point is narrower: when negative outcomes are largely unobserved, calibrating the false positive rate remains inherently difficult, since false positives are defined with respect to those missing negative outcomes. In this sense, relying on UQ alone is unlikely to fully replace the role of access to negative behavior data. Furthermore, we believe that such data can make UQ more effective by providing a stronger basis for identifying, calibrating, and reducing false positives.

---

> > ### Author Rebuttal · Reviewer_Gbia · 2026-04-02
> >
> > Thank you for answering my questions. I have no more concerns and will raise my score accordingly.

---

### Official Review · Reviewer_WwNz · 2026-03-13

**Significance:** 3
**Argument Clarity:** 3
**Rating:** 5
**Confidence:** 4

**Questions:**

Please see the weakness. The discussion and evidence in the paper is good, however, it could be further improved to cover more range of tasks, domains, and do thorough analysis and coverage of failure types.

**Alternative Views Section:**

Yes

**Compliance With Llm Reviewing Policy A Conservative:**

Affirmed.

**Discussion Potential:**

3

**Final Justification:**

I appreciate the positive engagement of the authors during the rebuttal. I will remain my positive rating and it would be good for the authors to update the related discussion in the revised version.

**Paper Summary:**

The paper states the position: embodied reaward models require failture data during their training. The authors argue that today's embodied reward models that trained on purely successful rollout data systematically over-reward behaviors that human would penalize. The authors conducted experimets with three different setups with real robot data to show that the reward model fail as task complexity increases, and the simple injection of failure data would strengthen the capability the reward models. At last, the authors call the community's action on releasing failure data, enhancing evaluation system, improving failure data synthesis and new embodied rewarding benchmarks.

**Position:**

Yes

**Position In Title:**

Yes

**Related Work:**

3

**Strengths And Weaknesses:**

Strengths
1. The paper does adequate experiments with preference-based reward models trained with synthetic negatives, zero-shot VLM rewards, and fine-tuned VLM rewards. These impirical results and quantitative numbers support the stated position.
2. The topic itself is emerging and requiring to those who doing embodied tasks in ICML community. Training on only correct or deliberately messed failure data will largely limit the embodied reward model or more the embodied foundation models.
3. The authors call several actions related to the stated position, \ie, release curated robot failure dataset, new methods for synthesizing bad robot data, more deconteralized and physical evalaution systems, new benchmarks for evaluating general-purpose embodied reward models.

Weakness
1. The current experiments dont include the soft object, long-horizon robot tasks, and navigation tasks. Here is why this would matter, it is hard to define "bad" when manipulating soft object - rigid object manipluation can be easily segmented over task completion, while soft object is not, \eg, folding blanket. For long-horizon robot tasks, human intervention would be more possible, what role will failure data play? Lastly, for navigation tasks, since the operaion area is totally different from robotic manipulation, what kind of failture data this domain want, will these two domains benifit each other by involving failture data?
2. Lack of detailed discussion over failture types. Questions like what kind of failure will affect most the embodied model training, what kind of failure should still be withheld are better to be answered.

**Support:**

3

---

> ### Author Rebuttal · Authors · 2026-03-31
>
> We thank the reviewer for the thoughtful feedback. We are glad that the reviewer finds the topic timely and important for the embodied AI community and that the experiments and discussion support the paper’s central position. We believe the suggested revisions will make the paper clearer and stronger.
>
> ***Q1. The current experiments don't include the soft object, long-horizon robot tasks, and navigation tasks.***
>
> A1. We would like to clarify that, although our task category names did not make this fully explicit, our experimental tasks do cover soft-object manipulation (e.g., folding cloth) as well as some longer-horizon behaviors (e.g., multi-step tool use) as shown in the website (https://sites.google.com/view/icml-2026-rebuttal-paper-782/home). We appreciate the reviewer pointing out that this scope should be stated more clearly, and we will add this clarification to Section 3.2. We also agree that it is often harder to define “bad” behavior in soft-object manipulation than in more rigid manipulation settings, since failures may be less cleanly segmentable by task completion alone. We believe this is precisely one reason why human-calibrated failure data can be especially valuable: in these settings, the notion of failure is often more nuanced and preference-dependent, making it harder to capture using only simple progress or end-state signals.
>
> ***Q2. For navigation tasks, since the operation area is totally different from robotic manipulation, what kind of failure data does this domain want? Will these two domains benefit each other by involving failure data?***
>
> A2. As for navigation tasks, we agree that navigation introduces a meaningfully different operating domain from manipulation, and we did not explicitly study navigation in the current paper. We think that the broader principle is still likely to carry over: navigation reward models may also require exposure to failure data that reflects the kinds of behaviors humans care about, such as safety violations, rule violations, and socially inappropriate motions. At the same time, we hypothesize that some failure concepts from the manipulation domain may still be beneficial to navigation reward models at a higher level, for example by exposing broader patterns such as unsafe shortcuts or collisions. However, additional navigation-specific failure data can further help improve navigation reward models. We appreciate this suggestion and will revise the paper to make the current scope clearer and to discuss navigation as an important extension direction.
>
> ***Q3. For long-horizon robot tasks, human intervention would be more possible, what role will failure data play?***
>
> A3. In the current experiments, we do not explicitly consider interventions. We agree, however, that human intervention introduces an important additional dimension. In particular, intervention can play several important roles, such as marking the onset of failure or near-failure, identifying portions of a trajectory where behavior begins to deviate from what is acceptable, and serving as a useful form of feedback for reward learning. We will revise the paper to clarify the current scope of our study in the experimental section and to discuss the role of intervention more explicitly in Section 7.
>
> ***Q4. Questions like what kind of failure will affect the most embodied model training, what kind of failure should still be withheld are better to be answered.***
>
> A4. We would like to thank the reviewer for this helpful question. We agree that a more detailed discussion of desirable failure types would strengthen the paper since not all failures are equally important. In particular, we believe that informative failures are those that are safety-critical or otherwise strongly misaligned with user preferences, especially when the behavior may still appear to make progress or even resemble task completion on the surface. These failures are especially valuable for training embodied reward models because they expose cases where reward models may become overly optimistic despite clear disagreement with human judgment. At the same time, we agree that some failure data may need to be withheld or handled more carefully, especially when collecting or releasing such data would require deliberately inducing unsafe behaviors outside controlled settings or when the data raises privacy, ethical, or deployment-risk concerns. We appreciate this suggestion and will edit Action #1 to include the discussions of failure types and the need for careful restrictions or controlled-access handling.

---

> > ### Author Rebuttal · Reviewer_WwNz · 2026-04-01
> >
> > Thanks for the rebuttal.

---

### Official Review · Reviewer_zF5r · 2026-03-15

**Significance:** 3
**Argument Clarity:** 4
**Rating:** 5
**Confidence:** 4

**Questions:**

Q1) From your findings, what kind of bad behavior is the most crucial for reliable reward modeling?

**Alternative Views Section:**

Yes

**Compliance With Llm Reviewing Policy A Conservative:**

Affirmed.

**Discussion Potential:**

4

**Final Justification:**

My main concerns have been addressed by the rebuttal. Based the comments from other reviewers, I choose to maintain my initial positive score.

**Paper Summary:**

The paper presents a position that good embodied reward models need bad behavior data. The authors conduct an analysis using preference-based reward models (ReWind), zero-shot visual language model (VLM) reward models (GVL), and fine-tuned VLM reward models (Dopamine) to demonstrate the limitations of existing reward modeling methods. The benefits of incorporating bad robot data are further validated with GVL. Based on their findings, the authors advocate for the release of large-scale failure datasets, the development of synthetic generation methods, more decentralized physical evaluation systems, and new benchmarks grounded in real human stakeholder values.

**Position:**

Yes

**Position In Title:**

Yes

**Related Work:**

4

**Strengths And Weaknesses:**

S1) The paper comprehensively covers all representative reward modeling methods.

S2) The position is supported by preliminary experimental results and analysis.

S3) Typical alternative views are also presented and discussed.

W1) The stated position is somewhat apparent. It would be more beneficial if the paper analyzed and discussed which types of bad data the community should prioritize first, and what would be the most promising approach to collecting those data at a relatively low cost while maximizing their returns.

**Support:**

3

---

> ### Author Rebuttal · Authors · 2026-03-31
>
> We would like to thank the reviewer for taking the time to review our paper and provide valuable feedback. We are glad that the reviewer finds our investigation thorough and that our position is supported by the experimental results and analysis.
>
> ***Q1. From your findings, what kind of bad behavior is the most crucial for reliable reward modeling?***
>
> A1. We thank the reviewer for this important question. We agree that not all failures are equally informative, and that identifying the most valuable kinds of bad behavior for reward modeling remains an important problem that deserves further study. From our experiments, failures that are safety-critical or otherwise strongly misaligned with user preferences, especially in cases where the behavior may still appear to make progress or even resemble task completion on the surface. We provide several qualitative examples in the paper that illustrate this pattern in Figure 2. More broadly, we also hypothesize that failures arising from long-horizon causality, failures that require temporal information to evaluate correctly, and failures on which current models exhibit high uncertainty may be important for reward modeling.
>
>
> ***Q2. What would be the most promising approach to collecting those data at a relatively low cost while maximizing their returns?***
>
> A2. We would like to thank the reviewer for raising this question! We agree that identifying cost-effective strategies for collecting high-value bad behavior data is a very important open problem, and this is also one of the main directions that our position paper aims to encourage the community to study more systematically in the Call to Action section.
>
> In particular, we believe one promising first step is to develop protocols and standards for leveraging naturally occurring bad behavior that is already produced by existing robotic systems. Our goal is not necessarily to induce new dangerous behavior, although such collection may be possible in carefully controlled lab or field settings, but rather to stop discarding negative data that already arises during routine operation, debugging, and deployment. We believe this is a particularly promising direction because such data is already being generated in practice yet is often filtered out or left uncurated, despite its potential to provide highly valuable supervision for improving reward models.
>
> Furthermore, synthetic bad behavior generation via high-fidelity simulation could complement naturally occurring failures by expanding coverage to rare, diverse, or difficult-to-observe failure modes. In particular, such simulators could enable targeted red-teaming of policies or reward models, making it possible to actively mine failures that are maximally informative while reducing the need to collect all such failures solely through real-world deployment.

---

> > ### Author Rebuttal · Reviewer_zF5r · 2026-04-02
> >
> > I appreciate the authors' efforts in answering my questions. I will maintain my positive rating.

---

### Decision · Program_Chairs · 2026-04-30

**Decision:**

Accept (spotlight)

**Comment:**

This is a strong submission. The position is clearly stated and directly supported by evidence, the alternative views are substantively engaged, the related work is comprehensive, and the call to action is concrete. All three reviewers recommended acceptance, and all three had their concerns resolved in the rebuttal. This is a strong signal.

There are some limitations -- the failure taxonomy is underdeveloped and could be improved, there is incomplete coverage of the task space (soft-object manipulation, navigation, and long-horizon tasks are not fully covered). The rebuttal also clarified an important nuance that should be incorporated into the paper: the claim is not that VLM-based reward models have never seen any failure data, but that generic pretraining is an imperfect substitute for embodied, human-calibrated negative supervision. This distinction matters for the paper's argument to be precise rather than overreaching.

The recommended revisions for the camera-ready version are: clarify the task coverage in Section 3.2 to make the soft-object and multi-step scope explicit; add the failure taxonomy discussion from the rebuttal to the call to action section; sharpen the VLM argument to reflect the nuance about pretraining versus embodied-specific negative supervision; and discuss navigation and long-horizon tasks as extension directions.